# Experimental Investigation on Dynamic Tensile Behaviors of Engineered Cementitious Composites Reinforced with Steel Grid and Fibers

**DOI:** 10.3390/ma14227042

**Published:** 2021-11-20

**Authors:** Liang Li, Hongwei Wang, Jun Wu, Shutao Li, Wenjie Wu

**Affiliations:** 1Key Laboratory of Urban Security and Disaster Engineering, Beijing University of Technology, Ministry of Education, Beijing 100124, China; 18811169721@163.com (H.W.); wuwj05@cifi.com.cn (W.W.); 2School of Urban Railway Transportation, Shanghai University of Engineering Science, Shanghai 201620, China; cvewujun@sues.edu.cn; 3Institute of Defense Engineering, AMS, PLA, Beijing 100036, China

**Keywords:** engineered cementitious composites (ECC), dynamic tensile behavior, PVA fiber, KEVLAR fiber, steel grid, strain rate

## Abstract

Engineered cementitious composites (ECC) used as runway pavement material may suffer different strain rate loads such as aircraft taxiing, earthquakes, crash impacts, or blasts. In this paper, the dynamic tensile behaviors of the steel grid-polyvinyl alcohol (PVA) fiber and KEVLAR fiber-reinforced ECC were investigated by dynamic tensile tests at medium strain rates. The mixture was designed with different volume fractions of fibers and layer numbers of steel grids to explore the reinforcement effectiveness on the dynamic performance of the ECC. The volume fractions of these two types of fibers were 0%, 0.5%, 1%, 1.5%, and 2% of the ECC matrix, respectively. The layer numbers of the steel grid were 0, 1, and 2. The dynamic tensile behaviors of the PVA fiber and the KEVLAR fiber-reinforced ECC were also compared. The experimental results indicate that under dynamic tensile loads, the PVA-ECC reveals a ductile and multi-cracking failure behavior, and the KEVLAR-ECC displays a brittle failure behavior. The addition of the PVA fiber and the KEVLAR fiber can improve the tensile peak stress of the ECC matrix. For the specimens A0.5, A1, A1.5, and A2.0, the peak stress increases by 84.3%, 149.4%, 209.6%, and 237.3%, respectively, compared to the matrix specimen. For the specimens K0.5, K1, K1.5, and K2, the peak stress increases by about 72.3%, 147.0%, 195.2%, and 263.9%, respectively, compared to the matrix specimen. The optimum fiber volume content is 1.5% for the PVA-ECC and the KEVLAR-ECC. The KEVLAR-ECC can supply a higher tensile strength than the PVA-ECC, but the PVA-ECC reveals more prominent deformation capacity and energy dissipation performance than the KEVLAR-ECC. Embedding steel grid can improve the tensile peak stress and the energy dissipation of the ECC matrix. For the strain rate of 10^−3^ s^−1^, the peak stress of the A0.5S1 and A0.5S2 specimens increases by about 49.1% and 105.7% compared to the A0.5 specimen, and the peak stress of the K0.5S1 and K0.5S2 specimens increases by about 61.5% and 95.8%, respectively, compared to the K0.5 specimen.

## 1. Introduction

As an infrastructure airport runway might experience dynamic loads of different strain rates, such as aircraft taxiing, earthquakes, crash impacts, or blasts, in its service life, with potentially catastrophic consequences. Concrete material has been widely used in airport runways because of its good mechanical properties, including low surface deformation and high stiffness [1,2,3]. However, there is still a vital issue concerning the improvement of concrete ductility and toughness. Studies have confirmed that incorporating fiber into concrete can provide a high ductility material [4]. The better performance of fiber-reinforced concrete can even be maintained under different strain rate loads [5,6,7,8,9,10,11,12]. The energy absorption capacity and toughness of concrete can especially be enhanced by adding more fibers [13,14,15,16,17,18,19].

Compared to ordinary fiber-reinforced concrete, engineered cementitious composites (ECC) were proposed as airport runway pavement due to its crack control capability, energy dissipation, and superior ductility [20,21,22,23,24,25]. The crack control capabilities of ECC are due to its micro fracture mechanics. The high energy dissipation performance of ECC are achieved by microcracks in the strain hardening stage [26,27]. The superior ductility mainly focuses on its good tensile deformation ability, with a tensile strain range of 3–7% [28].

Many researchers have further analyzed the tensile properties of ECC reinforced by various types of fibers [29]. The fibers mainly include carbon fiber, steel fiber, polymer fiber, basalt fiber, polyethylene (PE) fiber, and polyvinyl alcohol (PVA) fiber [30,31,32,33]. Incorporating these fibers individually or as a hybrid has different reinforcement effectiveness in the tensile behaviors of ECC. Vantadori et al. [34] discussed the fracture toughness of a unidirectional glass fiber-reinforced plastic (GFRP). Khandelwal et al. [35] confirmed that the mechanical performance of composites improves significantly as the dosage of the blast fiber increases. Rostami et al. [36] proposed the addition of nonindented and hydrophobic fibers, which resulted in a 25% increase in the splitting tensile strength of ECC. Wei et al. [37] presented a study adding 0.1 vol% steel fibers and 1.9 vol% PVA fibers, which resulted in an 11.8% increase in the load after the cracking of the ultra-high performance cementitious composite. Wang et al. [38] published a study incorporating 2.0 vol% of PE fiber, which made the tensile strain of the ECC achieve 11.99%. Özkan et al. [39] confirmed a good bond strength between the PVA fiber and ECC, and 2.0 vol% of the PVA fiber made the tensile strain of the ECC achieve 3.95%. Adesina [40] found that chopped basalt fibers and PVA fibers at a dosage of 2% resulted in an 81.8% and 96.4% increase in the tensile strength of ECC. Yu et al. [41] denoted that PE fiber can increase the fracture energy of ECC from 8.68 kJ/m^−2^ to 15.43 kJ/m^−2^. Yoo et al. [42] proposed that the modification of PE fiber can further improve the tensile strength of PE-ECC. Liu et al. [43] optimized the fiber volume fraction to enhance the tensile strength and the cracking pattern of ECC and found 1.5% of PE fibers and 0.68% of steel fibers were optimal. Huang et al. [44] discussed the hybrid effect of PE fiber and steel fiber on the ultra-high strength ECC and found 2.0 vol% of PE fiber and 1.0 vol% of steel fiber improved the tensile strain to 5.2%.

Understanding the tensile behaviors of ECC with different strain rates is vital to the design of airport runways under aircraft taxiing, earthquakes, crash impacts, or blast loads. The strain rate for aircraft taxiing and landing is 10^−2^–1 s^−1^; for earthquakes, it is about 10^−3^–10^−1^ s^−1^; for crash impacts, it is 1–50 s^−1^; and for blast loads, it is beyond 10^2^ s^−1^ [45]. The rate sensitivity of ECC can be influenced by the load rates and fiber types. Silva et al. [46] found that a low strain rate (10^−2^ s^−1^) will reduce the strain capacity, while a high strain rate (10–50 s^−1^) will improve the strain capacity of ECC. The ultimate tensile strength of ECC can be increased by 55% at high strain rates [47]. Tran and Kim [48] presented the rate sensitivities of ECC due to the reinforcement effectiveness of various fiber types. The dynamic increasing factor (DIF) of the tensile strength of the ECC reinforced by aramid fiber is 6.5, by polypropylene (PP) fiber it is 9.1, and by carbon fiber it is 6.7 at high strain rate [49]. The dynamic increasing factor (DIF) of the tensile strength of ECC reinforced by PVA fiber is 2.9 and by steel fiber it is 2.68 at middle strain rate [50].

Incorporating a textile grid in the ECC matrix can provide an individual structural material for the airport runway pavement. Daskiran et al. [51] and Ferrara et al. [52] have confirmed that textile-reinforced ECC has good mechanical and durability properties [53]. These properties are attributed to the high tensile strength of the textiles [54]. Textile grid-reinforced ECC also has a high energy absorption because of the typical strain hardening stage in the tensile tests [55,56]. Besides, the textile grid can improve the bearing capacity and the life expectancy of ECC, as they reduce cracking and damage [57]. Different textile grids include polyparaphenylene benzobisoxazole (PBO), carbon, basalt, glass, and steel grid [58]. The tensile behaviors of the ECC with these different textile grids have been researched. Gemeel et al. [59,60,61] discussed the tensile behavior of ECC embedded with basalt grid and found it has a more visible multiple crack pattern compared to the normal ECC. Dong et al. [62] proposed the tensile peak strength of ECC can be greatly enhanced by adding PVA fibers and glass textiles. Deng et al. [63] found that three layers of carbon grids can improve the peak stress of PVA-ECC by 130%. The flexural load and the toughness of the carbon textile-reinforced concrete were increased more than 380% and 820%, respectively, by a decrease in the carbon woven textiles’ mesh size from 20 to 2 mm [64]. Zhu et al. [65] presented that carbon grids can dominate the tensile behavior of ECC and improve its crack resistance. Kalaimathi et al. [66,67] found that the inclusion of textile fibers provides much more performance in tensile, toughness, flexural, energy absorption capacity, and ductility in ECC composites.

The available literatures focused on the tensile behaviors of ECC with steel grid are very limited when compared to other textile grids. Li et al. [68] investigated the tensile behaviors of ECC reinforced with steel grid and fibers under quasi-static tests. Padalu et al. [69] experimentally compared the tensile properties of 75 welded wire mesh and basalt fiber mesh-reinforced cementitious matrix composites. The experiment results showed the cementitious matrix composites reinforced by steel grids had better strength under tensile loads. Embedding steel grid into the engineered cementitious composites is similar to embedding a steel bar into concrete material. More information is needed to further discuss the tensile behaviors of the steel grid-ECC. It is still questionable whether the better tensile strength and strain-hardening behavior of the ECC with steel grid can even be maintained under different strain rate loads.

In this paper, the dynamic tensile behaviors of the steel grid fiber-reinforced ECC were studied. A series of tensile tests were conducted at medium strain rates to provide a referenced design instruction for airport runways under aircraft taxiing and landing. The test variations included two types of fiber (PVA fiber and KEVLAR fiber), the volume content of fiber (0%, 0.5%, 1%, 1.5%, 2%), the number of steel grid layers (0, 1, 2), and the strain rates (10^−4^ s^−1^, 10^−3^ s^−1^ and 10^−2^ s^−1^). The cracks development and failure patterns of specimens were recorded and discussed. The reinforcement effectiveness of the fiber volume content and the number of steel grid layers of the steel grid fiber-reinforced ECC was analyzed and discussed. The dynamic tensile performances of the PVA fiber and the KEVLAR fiber-reinforced ECC at various strain rates were also compared. The optimal fiber volume content for the energy dissipation was presented.

## 2. Experimental Program

### 2.1. Raw Materials and Mixture Proportion

The raw materials used in the ECC matrix include cement (P.O 42.5), silica fume, water, and superplasticizer. The properties of cement (P.O 42.5) are shown in Table 1 and Table 2. The content of SiO**_2_** in silica fume is 93.7%. The water reducing rate of the superplasticizer is 28%. The PVA fiber and the KEVLAR fiber were incorporated into the matrix as reinforcement material to achieve the ECC material. The properties of the PVA fiber and the KEVLAR fiber are given in detail in Table 3 and Table 4. The PVA fiber and the KEVLAR fiber with a length of 12 mm are shown in Figure 1. Steel grids were utilized with opening sizes of 12.8 mm × 12.8 mm, as shown in Figure 2. The wire diameter of the steel grid is 0.88 mm. The radial and latitudinal tensile strength of the steel grid is 478.18 MPa and 452.14 MPa, respectively. In the current study, the radial direction of the steel grid is the loading direction.

The mix proportions employed in the ECC matrix production are shown in Table 5. In the mix proportions, other components include the mass ratio to cement. The mixture was designed with different volume fractions of fibers and layer numbers of steel grids so to explore the reinforcement effectiveness on the dynamic performance of ECC. The volume fractions of these two types of fibers are 0%, 0.5%, 1%, 1.5%, and 2% of ECC matrix, respectively. The layer numbers of the steel grid are 0, 1, and 2, respectively.

### 2.2. Specimen Preparation and Test Set-Up

In the current study, a thin plate shape specimen of steel grid fiber-reinforced ECC was used. The length, width, and thickness of the ECC specimen is 300 mm, 75 mm, and 20 mm, respectively. Holes were reserved at each end of the specimen for connecting with the test fixture. Each end of the specimen was treated with reinforcement to prevent damage under loading. The inside of specimen end was reinforced by embedding three steel grid layers, as given in Figure 3. The outside surface of the specimen end was reinforced by a sticking steel plate, as given in Figure 4.

The preparation of the specimen mainly includes the following steps. Firstly, the steel grid layer was pre-embedded in the end of the self-developed steel mold. Secondly, the raw materials of the ECC matrix and fibers were mixed according to the design proportions. Dry powder ingredients, including cement and silica fume, were stirred in a mixer for 3 min, and the mixture of water and superplasticizer were put into the mixer and stirred for 2 min. Next, the fibers were manually added to the mixture and stirred for 3 min. Thirdly, fresh ECC was cast into the steel mold and placed on a vibrating table for consolidation. Then, the specimen surface was smoothed and was covered with plastic sheets to prevent the evaporation of moisture. Finally, the specimen was demolded after 24 h and placed in a standard curing room (temperature: 20 ± 0.5 **°**C, relative humidity: 95 ± 5%). After curing for 28 days, steel plates with holes were attached to the end of specimen as reinforcement.

The dynamic tensile tests were conducted by a universal testing machine produced by Germany Zwick/Roell Group, as shown in Figure 5. The maximum measure range of this testing machine is 100 kN. The specimen was connected with this test device through the pin shaft to ensure the axial tension. The tensile tests were performed at different strain rates of 10^−4^ s^−1^, 10^−3^ s^−1^, and 10^−2^ s^−1^. Two extensometers were attached on the specimen to measure the deformation, as given in Figure 6. The measure range of extensometer is 100 mm.

In the denotation of the specimen type, “M” stands for the matrix, “S” stands for the steel grid, “A” stands for the PVA fiber, and “K” stands for the KEVLAR fiber. Two numbers stand for the volume content of fiber and the layer number of steel grid, respectively. For example, “A0.5S2” stands for the ECC specimen reinforced with two layers of steel grid, and the volume content of the PVA fiber is 0.5%. The compositions of the various specimens used in the current experimental study are listed in Table 6 and Table 7. For every loading case, i.e., every combination of certain fiber volume content, the layer number of the steel grid, and the tensile strain rate, four specimens were tested, and the average value of the test results was used for the comparative analysis. A total of 180 PVA-ECC specimens and 180 KEVLAR-ECC specimens have been tested in the current study.

## 3. Results and Discussion

### 3.1. Dynamic Tensile Test Results of the Steel Grid-PVA-ECC

#### 3.1.1. Failure Patterns

The representative failure patterns of the steel grid-PVA-ECC specimens at a strain rate of 10^−3^ s^−1^ are presented in Figure 7. The failure patterns of all types of the steel grid-PVA-ECC specimens are exhibited in Appendix A (Figure A1 and Figure A2). The matrix specimens without the PVA fiber addition show a brittle failure behavior, and a main crack transfixes the specimen. With the addition of the PVA fiber, the specimens show a multi-cracking failure behavior, and microcracks emerge gradually on the surface of the specimen during the tensile loading course. The more obvious multi-cracking phenomenon can be observed with the increase of the volume content of the PVA fiber. This can be mainly attributed to the addition of the PVA fiber improving the toughness of the specimen under the dynamic tensile loading. The incorporation of the steel grid enhanced the tensile strength of the specimen but weakened its multi-cracking failure behavior. This may be because the steel grid suffered to the main tension after the specimen was cracked.

Figure 8 illustrates the cross-section microcosmic view of the damaged steel grid-PVA-ECC specimen ($\dot{\varepsilon }$ = 10^−3^ s^−1^) obtained by scanning electron microscope. It can be found that the PVA fiber distributed uniformly in the matrix. When the specimen was damaged, the PVA fiber was pulled out without fracture in the cracks. This means that during the cracking process of the specimen, the PVA fiber can present a bonding action in the matrix which dissipates the tensile energy.

The dynamic tensile test results at different tensile strain rates, including the peak stress, ultimate strain, and energy dissipation, are presented in Table 8. The ultimate strain is defined as that corresponding to 80% of the peak stress in the descending segment of the stress–strain curve. The energy dissipation is obtained from the integration of the stress–strain curve. The influencing factors of the dynamic tensile behaviors, including the volume content of fiber, the number of steel grid layers, and the strain rate, will be discussed respectively in the following sections in detail.

#### 3.1.2. Effect of the Volume Contents of PVA Fiber

Figure 9 presents a comparison of the stress–strain relationships of the PVA-ECC specimens with different fiber volume contents. Figure 10 shows the corresponding peak stress and the ultimate strain of the above specimens with different fiber volume contents. The peak stress of the PVA-ECC increased remarkably when compared to the ECC matrix. For the specimens A0.5, A1, A1.5, and A2.0, the peak stress increases by 84.3%, 149.4%, 209.6%, and 237.3% compared to the matrix specimen at a strain rate of 10^−4^ s^−1^, respectively. The specimen reveals more obvious strain hardening behavior under the dynamic tensile loading. Moreover, the stress–strain curve appears at a longer strain hardening stage. This indicates that the fibers play a noticeable bonding role in the ECC matrix and enhance its deformation capacity.

#### 3.1.3. Effect of the Number of Steel Grid Layer

Figure 11 presents the effect of the steel grid layers on the tensile behaviors of the specimens in the groups A0.5 and A1. It can be found that the peak stress of the steel grid-ECC have increased when compared to the ECC without the steel grid. For the group A0.5 at strain rate 10^−3^ s^−1^, the peak stress of the A0.5S1 and A0.5S2 specimens increases by about 49.1% and 105.7% compared to the A0.5 specimen, respectively. For the group A1, the peak stress of the A1S1 and A1S2 specimens increases by about 29.6% and 75.4% compared with A1 specimen, respectively. The stress–strain curves of the steel grid-ECC specimens have obvious strain hardening stages. Figure 12 reveals the dynamic tensile energy absorption of the different types of ECC specimens. It is illustrated that specimens with two steel grid layers, such as A1S2, A1.5S2, and A2S2, have a higher energy dissipation than the other types of specimens. This may be because, after the cracking of the specimen, the steel grid is not broken up immediately and can still supply tensile resistance and an energy consumption effect.

#### 3.1.4. Effect of Tensile Strain Rate

The comparison of the dynamic tensile stress–strain relationships of the steel grid-PVA-ECC specimens at various tensile strain rates is presented in Figure 13. Figure 14 shows the variation of the peak stress and the ultimate strain of the steel grid-PVA-ECC specimens at various tensile strain rates. As a whole, the peak stress and the ultimate strain of the ECC specimens will be enhanced with the increase of the tensile strain rate. Compared to the test results at a strain rate of 10^−4^ s^−1^, the peak stress of the A1S2 specimen increases by about 5.51% and 9.52% at strain rate of 10^−3^ s^−1^ and 10^−2^ s^−1^, respectively. The peak stress of the A1.5 specimen increases by about 10.1% and 18.3% at strain rate of 10^−3^ s^−1^ and 10^−2^ s^−1^, respectively. Compared to specimens at a strain rate of 10^−4^ s^−1^, the ultimate strain of the A1S2 specimen and A1.5 specimen has an increase of about 13.9% and 33.3% at strain rate of 10^−2^ s^−1^ and 10^−3^ s^−1^, respectively.

### 3.2. Dynamic Tensile Test Results of the Steel Grid-KEVLAR-ECC

#### 3.2.1. Failure Patterns

The representative failure patterns of the steel grid-KEVLAR-ECC specimens at a strain rate of 10^−3^ s^−1^ are presented in Figure 15. The failure patterns of all types of the steel grid-KEVLAR-ECC specimens are exhibited in Appendix A. The brittle failure pattern can be observed for most specimens, and only a main crack appears and transfixes the specimen. The multi-crack phenomenon cannot be observed obviously even for the specimens with a relatively high volume of content of the KEVLAR fiber.

Figure 16 presents the microscopic view of the cross section of the damaged steel grid- KEVLAR-ECC specimen ($\dot{\varepsilon }$ = 10^−3^ s^−1^). It is found that the KEVLAR fibers are unevenly distributed in the ECC matrix, and some fiber agglomerations can be observed. There are two possible reasons for this result:(1)Some grease exists on the surface of the KEVLAR fiber. They can be cleaned out by alcohol. The KEVLAR fibers are agglomerated when mixing with alcohol.(2)When the specimen was fabricated, the fresh ECC mixtures were cast into steel molds. The steel grid needs to be embedded in the mold in advance. The space for the casting of the ECC mixture was limited. It was difficult for the ECC mixtures being cast into the bottom of steel mold. This leads to an inhomogeneous distribution of the KEVLAR fiber in the ECC matrix.

The dynamic tensile test results of the steel grid- KEVLAR-ECC, including the peak stress, the ultimate strain, and the energy dissipation are presented in Table 9. The influencing factors of the dynamic tensile behaviors, including the volume content of fiber, the number of steel grid layers, and the strain rate will be discussed respectively in the following sections in detail.

#### 3.2.2. Effect of the Volume Contents of KEVLAR Fiber

Figure 17 shows the stress–strain relationships of the KEVLAR-ECC specimens with various fiber volume contents. The peak stress and the ultimate strain of the KEVLAR-ECC have increased remarkably compared to the ECC matrix. For specimens K0.5, K1, K1.5, and K2, the peak stress increases by about 72.3%, 147.0%, 195.2%, and 263.9% compared with the matrix specimen at the strain rate of 10^−4^ s^−1^, respectively. The KEVLAR-ECC has better tensile deformation and energy dissipating capacity. The specimen K1.5 especially reveals the most obvious strain hardening behavior.

#### 3.2.3. Effect of the Number of Steel Grid Layer

Figure 18 presents the effect of the steel grid layers on the stress–strain curves in groups of K0.5 and K1 specimens. It is illustrated that the peak stress and the ultimate strain of the steel grid-ECC have increased compared to the ECC without the steel grid. For the group K0.5, the peak stress of the K0.5S1 and K0.5S2 specimens increases by about 61.5% and 95.8% compared to the specimen without steel grid at strain rate of 10^−4^ s^−1^, respectively. For the group K1, the peak stress of the K1S1 and K1S2 specimens increases by about 19.0% and 52.2% compared with the specimen without steel grid, respectively. The stress–strain curves of the steel grid-ECC specimens have an obvious strain hardening stage. Besides, the energy dissipating performance of ECC has been improved by the appending of the steel grid. Figure 19 reveals the dynamic tensile energy absorption of the steel grid-KEVLAR -ECC specimens. It is shown that the specimens with the steel grid, such as K1S2, K1.5S1, and K2S2, have a higher energy dissipation than the other types of specimens. The steel grid can maintain a considerable tensile resistance after the cracking of the ECC matrix.

#### 3.2.4. Effect of the Strain Rate

The comparison of the dynamic tensile stress–strain curves of the steel grid-KEVLAR-ECC specimens at different tensile strain rates is presented in Figure 20. Figure 21 shows the variation of the peak stress and the ultimate strain of the steel grid-KEVLAR-ECC specimens at various tensile strain rates. The peak stress of the ECC specimens is enhanced with the increase of the tensile strain rate. Compared to the test results at a strain rate of 10^−4^ s^−1^, the peak stress of K1.5 increases by about 18.3% and 27.2% at a strain rate of 10^−3^ s^−1^ and 10^−2^ s^−1^, respectively. The peak stress of K1.5S2 increases by about 16.7% and 27.3% at a strain rate of 10^−3^ s^−1^ and 10^−2^ s^−1^, respectively. Compared to the test results at a strain rate of 10^−4^ s^−1^, the ultimate strain of the K1.5 specimen increases by about 12.5% and 87.5% at a strain rate of 10^−3^ s^−1^ and 10^−2^ s^−1^, respectively. The ultimate strain of the K1.5S2 specimen increases by about 42.9% and 90.5% at a strain rate of 10^−3^ s^−1^ and 10^−2^ s^−1^, respectively.

### 3.3. Comparison of Dynamic Tensile Behaviors of PVA and KEVLAR Fiber Reinforced ECC

The comparisons of the stress–strain relationships of the PVA-ECC and the KEVLAR-ECC specimens with the same fiber volume content at different strain rates are presented in Figure 22, Figure 23 and Figure 24. The corresponding comparisons of the peak stress and the ultimate strain are illustrated in Figure 25. It is shown that the KEVLAR-ECC can present a higher peak stress than the PVA-ECC for most loading cases. For a tensile strain rate of 10^−3^ s^−1^, the peak stress of the K0.5, K1, K1.5, and K2 specimens have an increase of 113.2%, 13.8%, 9.9%, and 15.4%, respectively, compared to the A0.5, A1, A1.5, and A2 specimens. This may be because the KEVLAR fiber has a higher elastic modulus than the PVA fiber. The tensile modulus of the KEVLAR fiber is 70.5 GPa and the PVA fiber is 41 GPa. When experiencing the same tensile strain, the KEVLAR-ECC specimens can supply a higher tensile resistance. On the other hand, the comparison of the ultimate strain of the PVA-ECC and the KEVLAR-ECC specimens displays a complex case. The major trend is that the ultimate strain of the PVA-ECC is larger than that of the KEVLAR-ECC. This means that the PVA-ECC has better deformation capacity than the KEVLAR-ECC.

Figure 26 shows the comparison of the dynamic tensile energy absorption of the PVA and the KEVLAR fiber-reinforced ECC specimens. For most of the loading cases, the PVA fiber-reinforced ECC has a larger energy absorption value than that of the KEVLAR-ECC. The PVA-ECC displays a higher energy dissipation than that of the KEVLAR-ECC under the dynamic tensile loading. The energy dissipation capacity depends on the composite effect of the peak stress and the ultimate strain under the dynamic tensile loading. Besides, with the increase of the fiber addition, the energy dissipation capacity of the two types of ECCs does not reveal a monotonic variation, except for the case of the 2% PVA fiber at a strain rate of 10^−4^ s^−1^. This means that the energy dissipation capacity of the fiber-reinforced ECC is not directly proportional to the fiber volume fraction. Increasing the fiber content may lead to the uneven distribution of fibers and produce a defect in the ECC matrix. The defect will impact the tensile behaviors of the ECC and deteriorate its energy dissipation capacity. There is an optimum fiber volume content for the fiber-reinforced ECC from the consideration of its energy dissipation capacity. The optimum fiber volume content is 1.5% for the PVA-ECC and the KEVLAR-ECC. Especially for the PVA-ECC, the energy dissipation capacity has a remarkable enhancement when the fiber volume content is 1.5%. The energy absorption has an increase of 327.8%, 333.3%, and 73.7% at the strain rate of 10^−4^ s^−1^, 10^−3^ s^−1^, and 10^−2^ s^−1^, respectively, compared to the case when the fiber volume content is 1.0%.

## 4. Conclusions

In the current study, dynamic tensile tests at medium strain rates were carried out to investigate the dynamic tensile behaviors of reinforced ECC specimens. The applied tensile loads, strain rates, tensile stress, and the strain of the specimens were monitored during the tests. The crack development and failure patterns of the specimens were recorded and analyzed. The reinforcement effectiveness of the fiber and the steel grid layers were analyzed and discussed. The dynamic tensile behaviors of the PVA-ECC and the KEVLAR-ECC were also compared. It is concluded based on the experimental results that:(1)Under dynamic tensile loads, the PVA-ECC reveals a ductile and multi-cracking failure behavior. The more obvious multi-cracking phenomenon can be observed with the increase of the PVA fiber volume fraction. The multi-cracking failure behavior is weakened for the steel grid-PVA-ECC. On the whole, the steel grid-KEVLAR-ECC displays a brittle failure behavior under the dynamic tensile loading. The multi-crack phenomenon cannot be obviously observed even for the specimens with a relatively high-volume fraction of KEVLAR fiber.(2)The addition of PVA fiber and KEVLAR fiber can improve the tensile peak stress of the ECC matrix. The PVA-ECC reveals more obvious strain hardening behavior under dynamic tensile loading with the increase of fiber volume content. There is an optimum fiber volume content for the fiber-reinforced ECC from the consideration of the energy dissipation capacity. The optimum fiber volume content is 1.5% for the PVA-ECC and the KEVLAR-ECC.(3)The addition of steel grid can improve the tensile strength and the energy dissipation performance of the ECC matrix. With the increase of the number of steel grid layers, the tensile peak stress and the energy absorption are enhanced for both the PVA-ECC and the KEVLAR-ECC.(4)The steel grid-PVA-ECC and the steel grid-KEVLAR-ECC each display the strain rate effect during dynamic tensile loadings. On the whole, the peak stress and the ultimate strain of the two types of ECC will be enhanced with the increase of the tensile strain rate.(5)For the same fiber volume content, the KEVLAR-ECC can supply a higher tensile strength than can the PVA-ECC. However, the PVA-ECC reveals more prominent deformation capacity and energy dissipation performance than the KEVLAR-ECC.

## Figures and Tables

**Figure 1 materials-14-07042-f001:**
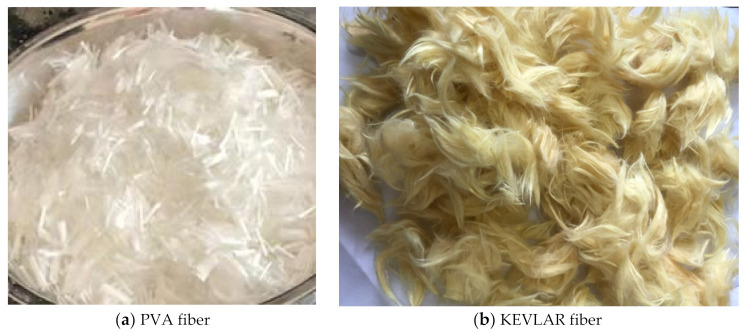
Two types of fibers.

**Figure 2 materials-14-07042-f002:**
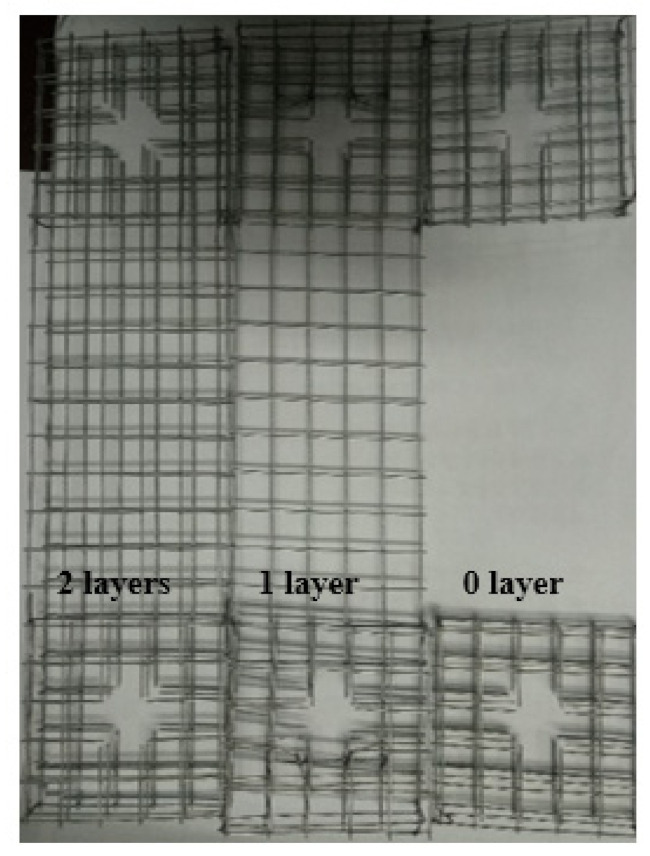
Steel grids used in the current study.

**Figure 3 materials-14-07042-f003:**
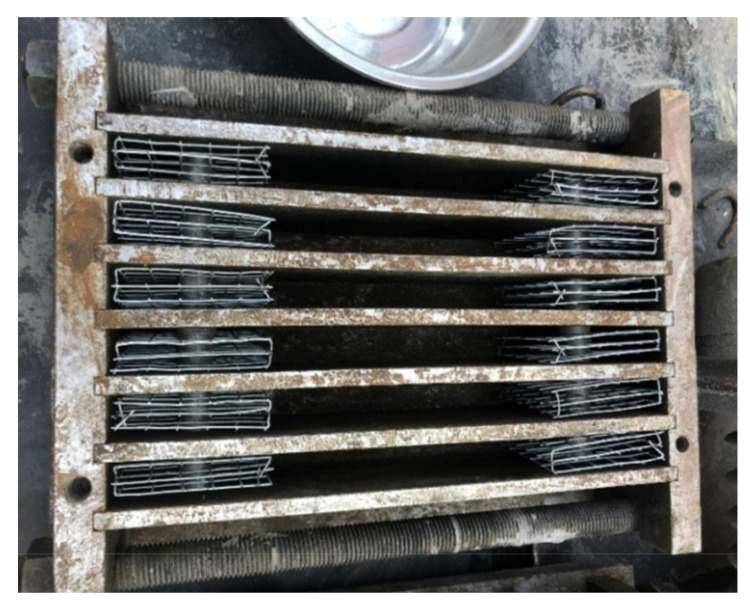
Steel mold for the specimens casting.

**Figure 4 materials-14-07042-f004:**
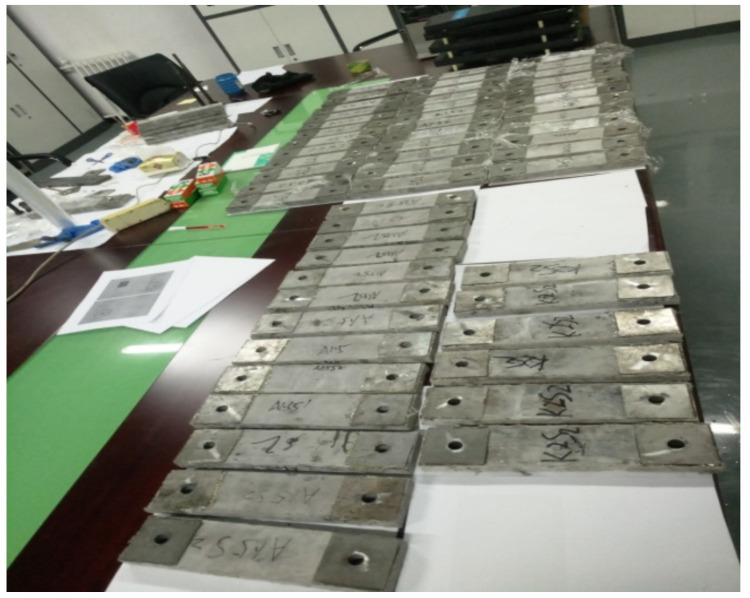
Specimens with the end reinforcement.

**Figure 5 materials-14-07042-f005:**
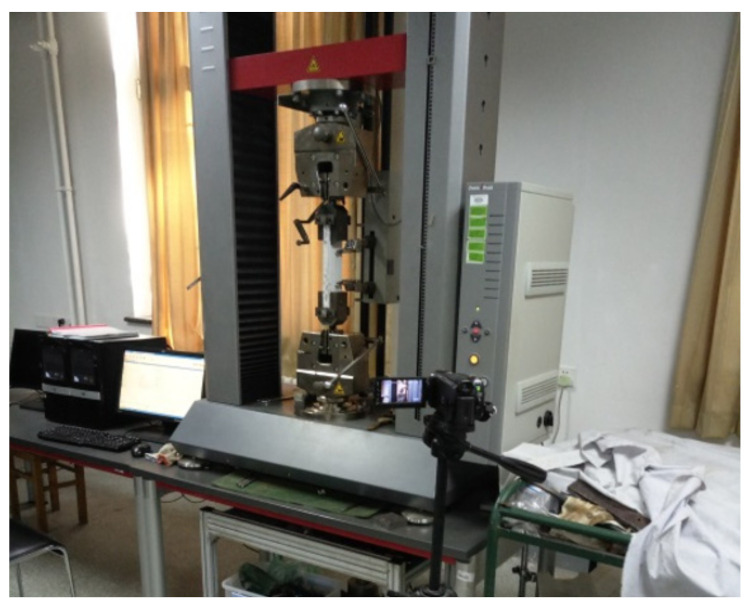
Z100 universal material testing machine.

**Figure 6 materials-14-07042-f006:**
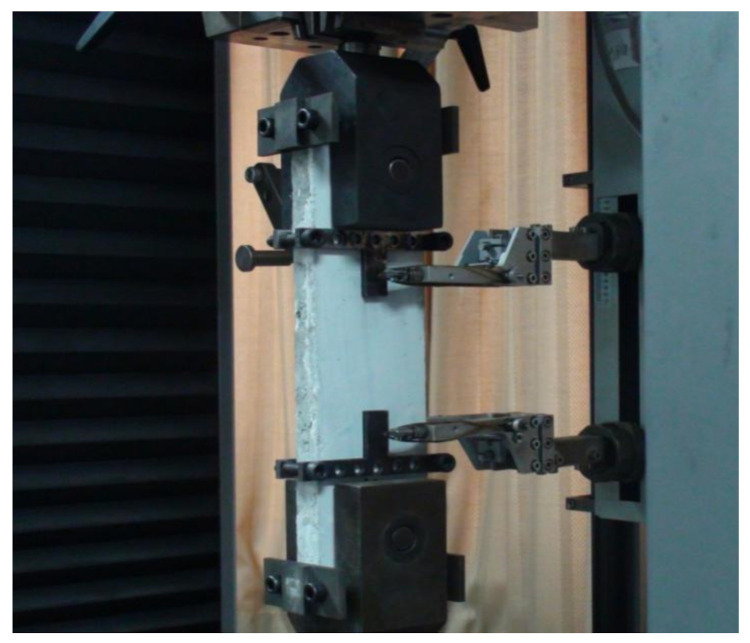
Specimen under the tensile loading.

**Figure 7 materials-14-07042-f007:**
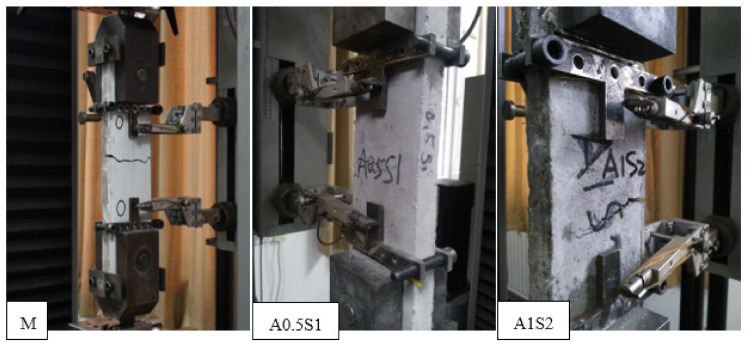
Representative cracking and failure of steel grid-PVA-ECC specimens.

**Figure 8 materials-14-07042-f008:**
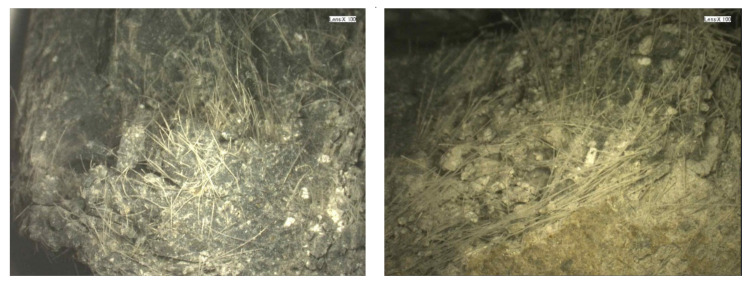
Microscopic view of cross section of damaged steel grid-PVA-ECC specimen ($\dot{\varepsilon }$ = 10^−3^ s^−1^).

**Figure 9 materials-14-07042-f009:**
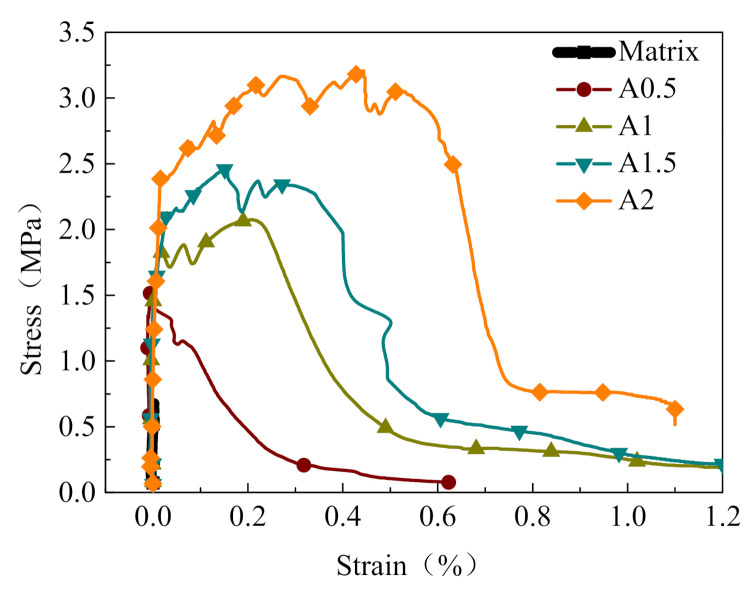
Stress–strain curves of PVA-ECC specimens ($\dot{\varepsilon }$ = 10^−4^ s^−1^).

**Figure 10 materials-14-07042-f010:**
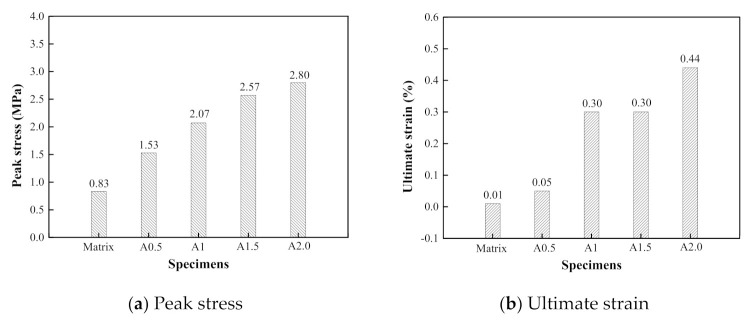
Effect of PVA fiber volume contents on peak stress and ultimate strain of ECC specimens ($\dot{\varepsilon }$ = 10^−4^ s^−1^).

**Figure 11 materials-14-07042-f011:**
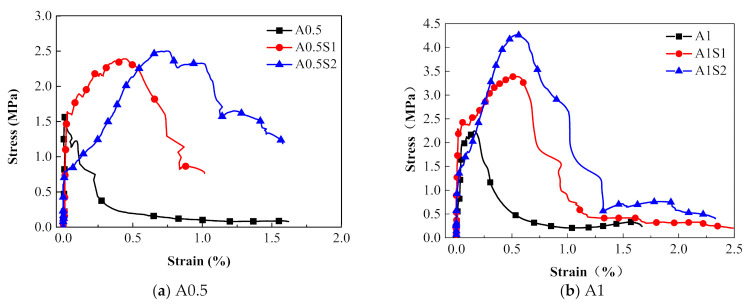
Effect of steel grid layers on stress–strain curves of ECC specimens ($\dot{\varepsilon }$ = 10^−3^ s^−1^).

**Figure 12 materials-14-07042-f012:**
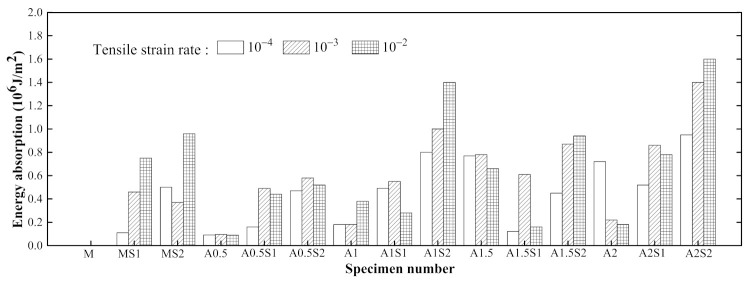
Comparison of dynamic tensile energy absorption of different types ECC specimens.

**Figure 13 materials-14-07042-f013:**
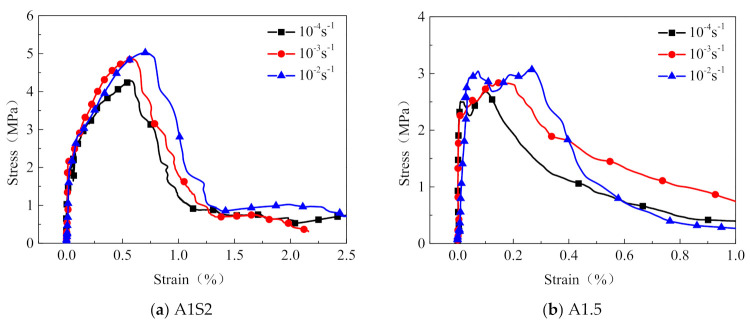
Dynamic tensile stress–strain curves of steel grid-PVA-ECC specimens at different strain rates.

**Figure 14 materials-14-07042-f014:**
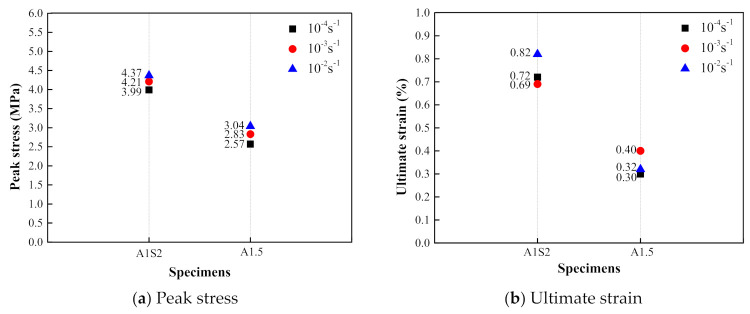
Variation of peak stress and ultimate strain of steel grid-PVA-ECC specimens at different strain rates.

**Figure 15 materials-14-07042-f015:**
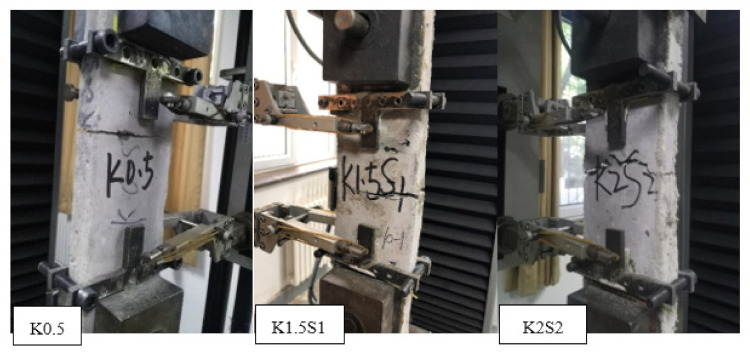
Representative **c**racking and failure of steel grid-KEVLAR-ECC specimens.

**Figure 16 materials-14-07042-f016:**
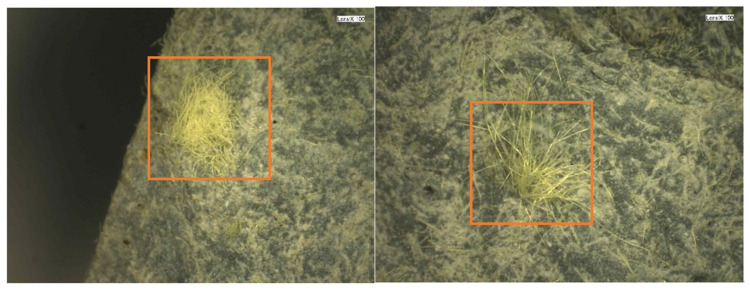
Microscopic view of cross section of damaged steel grid-KEVLAR-ECC specimen ($\dot{\varepsilon }$ = 10^−3^ s^−1^).

**Figure 17 materials-14-07042-f017:**
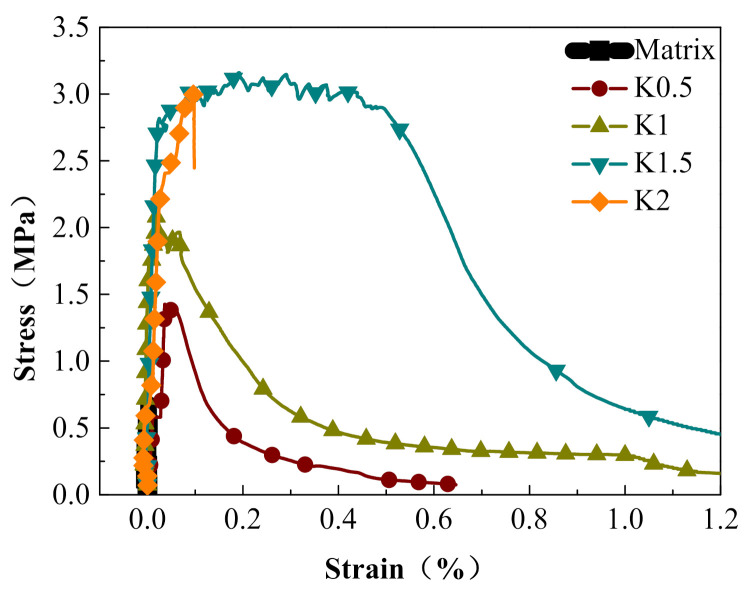
Stress–strain relationships of ECC specimens with various KEVLAR fiber volume contents ($\dot{\varepsilon }$ = 10^−4^ s^−1^).

**Figure 18 materials-14-07042-f018:**
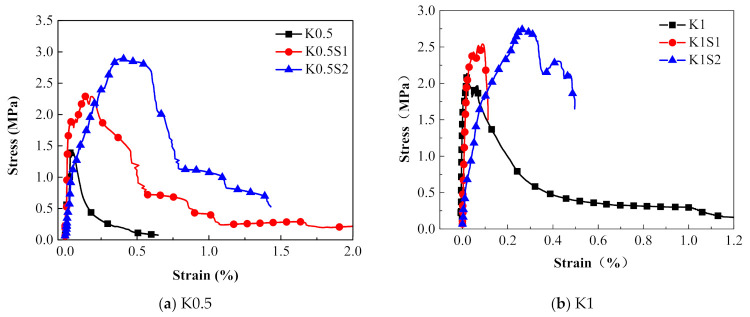
Effect of steel grid layers on stress–strain curves of ECC specimens ($\dot{\varepsilon }$ = 1 × 10^−4^ s^−1^).

**Figure 19 materials-14-07042-f019:**
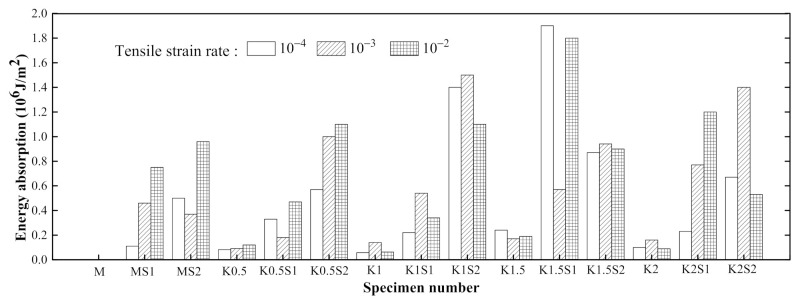
Comparison of dynamic tensile energy absorption of different types ECC specimens.

**Figure 20 materials-14-07042-f020:**
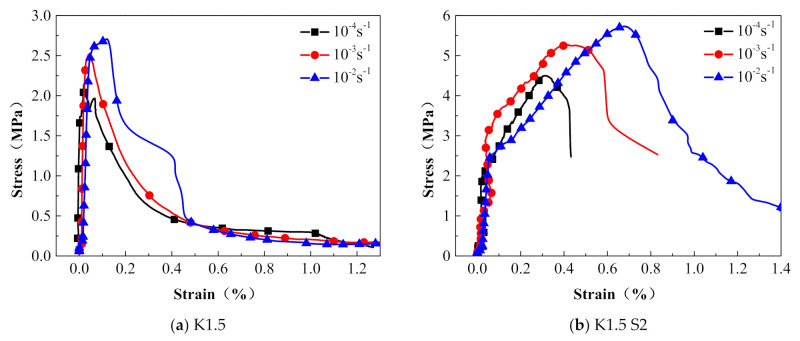
Stress–strain relationships of steel grid-KEVLAR-ECC specimens at different strain rates.

**Figure 21 materials-14-07042-f021:**
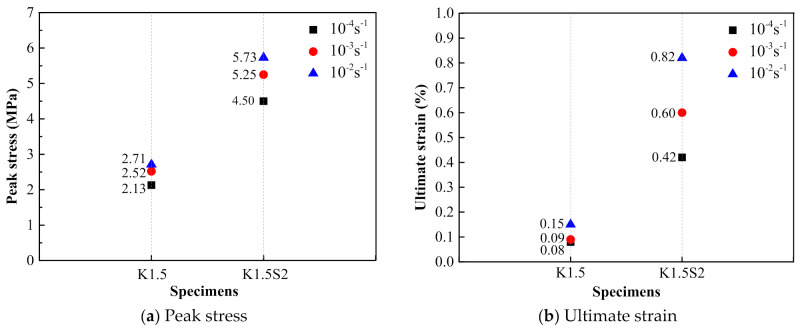
Variation of peak stress and ultimate strain of steel grid-KEVLAR-ECC specimens at various strain rates.

**Figure 22 materials-14-07042-f022:**
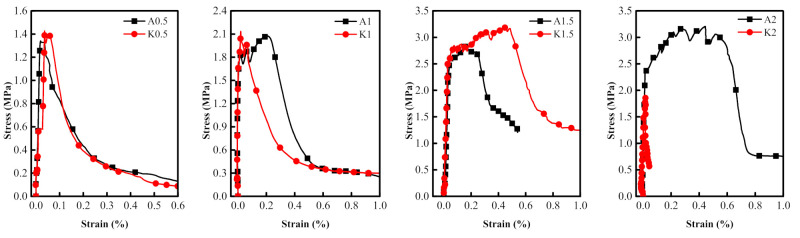
Stress–strain curves of PVA-ECC and KEVLAR-ECC specimens with same fiber volume content ($\dot{\varepsilon }$ = 10^−4^ s^−1^).

**Figure 23 materials-14-07042-f023:**
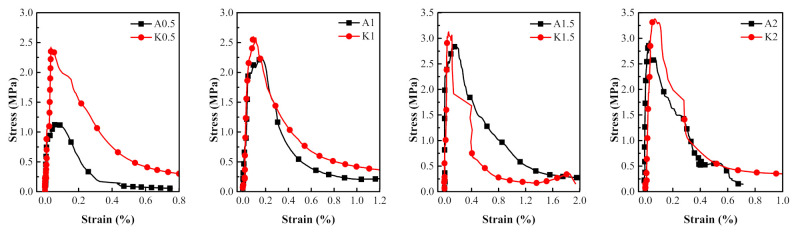
Stress–strain curves of PVA-ECC and KEVLAR-ECC specimens with same fiber volume content ($\dot{\varepsilon }$ = 10^−3^ s^−1^).

**Figure 24 materials-14-07042-f024:**
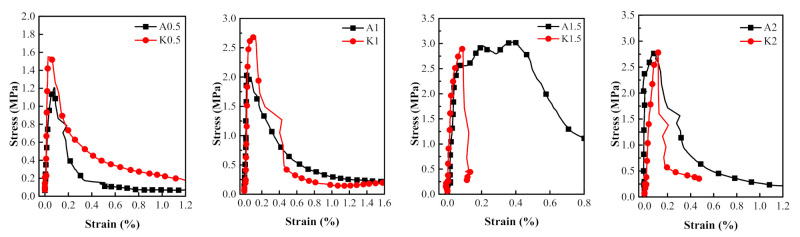
Stress–strain curves of PVA-ECC and KEVLAR-ECC specimens with same fiber volume content ($\dot{\varepsilon }$ = 10^−2^ s^−1^).

**Figure 25 materials-14-07042-f025:**
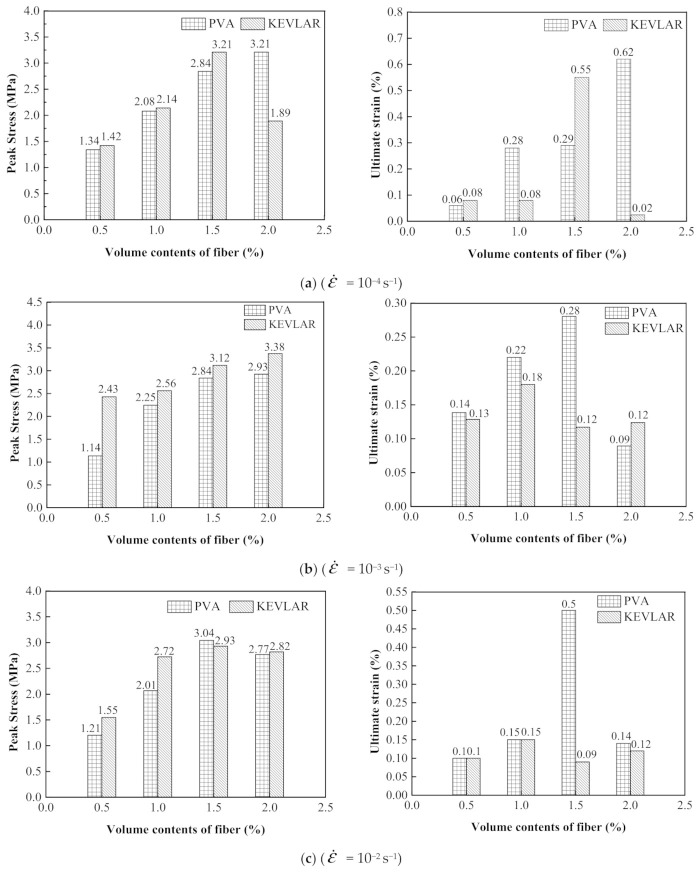
Comparison of peak stress and ultimate strain of PVA-ECC and KEVLAR-ECC specimens.

**Figure 26 materials-14-07042-f026:**
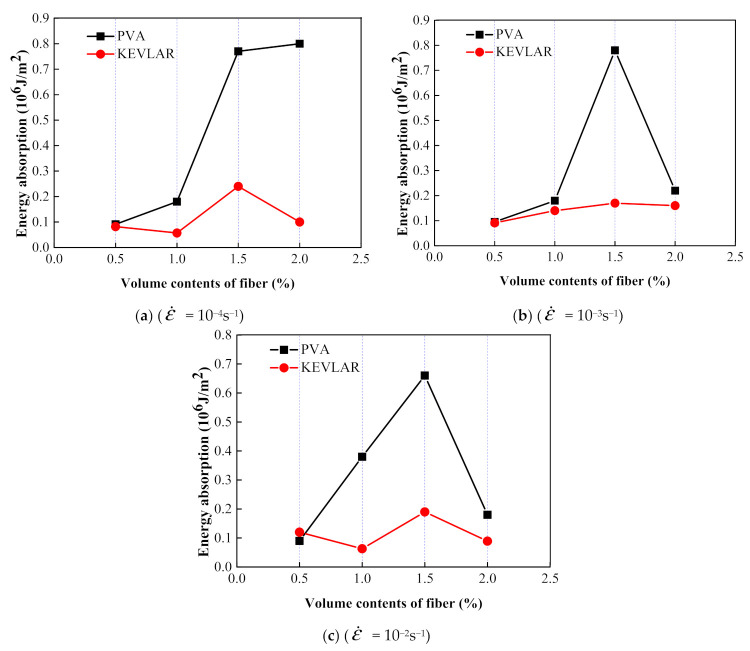
Comparison of dynamic tensile energy absorption of PVA-ECC and KEVLAR-ECC specimens.

**Table 1 materials-14-07042-t001:** Parameters of cement (P.O 42.5).

Specific Surface Area (m^2^/kg)	Initial Concreting Time (min)	Ultimate Concreting Time (min)	Compression Strength (3 days) (MPa)	Bending Strength (3 days) (MPa)
381	181	243	23.5	5.3

**Table 2 materials-14-07042-t002:** Chemical composition of cement (P.O 42.5).

Composition	SO_3_	MgO	Chloride ion	Loss on Ignition	Slag	Plaster
Content (%)	2.37	4.20	0.019	3.2	12	4

**Table 3 materials-14-07042-t003:** Properties of PVA fiber.

Diameter (μm)	Standard Length (mm)	Tensile Strength (MPa)	Elongation Ratio (%)	Young’s Modulus(GPa)	Density (g/cm^3^)
40	12	1560	6.5	41	1.3

**Table 4 materials-14-07042-t004:** Properties of KEVLAR fiber.

Breaking Strength (MPa)	Tensile Modulus (GPa)	Elongation Ratio (%)	Decomposing Temperature (°C)	Density (g/cm^3^)
2920	70.5	3.6	427–482	1.44

**Table 5 materials-14-07042-t005:** Mix ratio of ECC matrix (Mass Ratio).

Cement	Silica Fume	Water	Superplasticizer
1	0.11	0.3	0.013

**Table 6 materials-14-07042-t006:** Compositions of steel grid-PVA-ECC specimens.

Denotation	Volume Contents of Fiber	Layer Number of Steel Grid
M	0	0
MS1	0	1
MS2	0	2
A0.5	0.5%	0
A0.5S1	0.5%	1
A0.5S2	0.5%	2
A1	1.0%	0
A1S1	1.0%	1
A1S2	1.0%	2
A1.5	1.5%	0
A1.5S1	1.5%	1
A1.5S2	1.5%	2
A2	2.0%	0
A2S1	2.0%	1
A2S2	2.0%	2

**Table 7 materials-14-07042-t007:** Compositions of steel grid-KEVLAR-ECC specimens.

Denotation	Volume Contents of Fiber	Layer Number of Steel Grid
K0.5	0.5%	0
K0.5S1	0.5%	1
K0.5S2	0.5%	2
K1	1.0%	0
K1S1	1.0%	1
K1S2	1.0%	2
K1.5	1.5%	0
K1.5S1	1.5%	1
K1.5S2	1.5%	2
K2	2.0%	0
K2S1	2.0%	1
K2S2	2.0%	2

**Table 8 materials-14-07042-t008:** Dynamic tensile test results of the steel grid-PVA-ECC.

Specimen Type	Strain Rate (s^−1^)	Cracking Stress (MPa)	Peak Stress (MPa)	Ultimate Strain (%)	Energy Absorption (J/m^2^)
M	10^−4^	0.83	0.83	0.01	2.4 × 10^3^
	10^−3^	0.68	0.68	0.01	1.1 × 10^3^
	10^−2^	0.92	0.92	0.02	1.8 × 10^3^
MS1	10^−4^	1.05	1.37	1.4	1.1 × 10^5^
	10^−3^	1.03	1.32	0.73	4.6 × 10^5^
	10^−2^	0.75	1.35	1.23	7.5 × 10^5^
MS2	10^−4^	0.65	2.42	0.72	5.0 × 10^5^
	10^−3^	0.77	2.41	0.85	3.7 × 10^5^
	10^−2^	0.79	2.71	1.53	9.6 × 10^5^
A0.5	10^−4^	1.53	1.53	0.05	9.1 × 10^4^
	10^−3^	1.59	1.59	0.08	9.5 × 10^4^
	10^−2^	1.52	1.52	0.05	9.0 × 10^4^
A0.5S1	10^−4^	1.57	2.08	0.35	1.6 × 10^5^
	10^−3^	1.63	2.37	0.48	4.9 × 10^5^
	10^−2^	2.01	2.46	0.38	4.4 × 10^5^
A0.5S2	10^−4^	3.57	3.57	0.65	4.7 × 10^5^
	10^−3^	3.25	3.27	0.67	5.8 × 10^5^
	10^−2^	3.47	3.47	0.74	5.2 × 10^5^
A1	10^−4^	1.80	2.07	0.3	1.8 × 10^5^
	10^−3^	2.22	2.40	0.24	1.8 × 10^5^
	10^−2^	2.55	2.59	0.17	3.8 × 10^5^
A1S1	10^−4^	2.35	3.12	0.91	4.9 × 10^5^
	10^−3^	3.06	3.11	0.82	5.5 × 10^5^
	10^−2^	2.52	2.90	0.69	2.8 × 10^5^
A1S2	10^−4^	1.7	3.99	0.72	8.0 × 10^5^
	10^−3^	4.2	4.21	0.69	1.0 × 10^6^
	10^−2^	4.37	4.37	0.82	1.4 × 10^6^
A1.5	10^−4^	2.48	2.57	0.3	7.7 × 10^5^
	10^−3^	2.74	2.83	0.4	7.8 × 10^5^
	10^−2^	2.91	3.04	0.32	6.6 × 10^5^
A1.5S1	10^−4^	3.96	3.96	1	1.2 × 10^5^
	10^−3^	3.76	3.76	0.53	6.1 × 10^5^
	10^−2^	4.52	4.52	0.78	1.6 × 10^5^
A1.5S2	10^−4^	4.28	4.41	0.54	4.5 × 10^5^
	10^−3^	4.90	4.90	0.72	8.7 × 10^5^
	10^−2^	4.99	4.99	0.90	9.4 × 10^5^
A2	10^−4^	2.80	2.80	0.44	8.0 × 10^5^
	10^−3^	2.95	2.95	0.10	2.2 × 10^5^
	10^−2^	3.14	3.14	0.11	1.8 × 10^5^
A2S1	10^−4^	4.35	4.35	0.8	5.2 × 10^5^
	10^−3^	3.78	3.78	0.44	8.6 × 10^5^
	10^−2^	4.29	4.29	0.47	7.8 × 10^5^
A2S2	10^−4^	4.96	4.96	0.82	9.5 × 10^5^
	10^−3^	5.23	5.53	0.61	1.4 × 10^6^
	10^−2^	5.53	5.53	0.75	1.6 × 10^6^

**Table 9 materials-14-07042-t009:** Dynamic tensile test results of the steel grid-KEVLAR-ECC.

Number	Strain Rate (s^−1^)	Cracking Stress (MPa)	Peak Stress (MPa)	Ultimate Strain (%)	Energy Absorption (J/m^2^)
M	10^−4^	0.83	0.83	0.01	2.4 × 10^3^
	10^−3^	0.68	0.68	0.01	1.1 × 10^3^
	10^−2^	0.92	0.92	0.02	1.0 × 10^3^
MS1	10^−4^	1.05	1.37	1.4	1.1 × 10^5^
	10^−3^	1.03	1.32	0.73	4.6 × 10^5^
	10^−2^	0.75	1.35	1.23	7.5 × 10^5^
MS2	10^−4^	0.65	2.42	0.72	5.0 × 10^5^
	10^−3^	0.77	2.41	0.85	3.7 × 10^5^
	10^−2^	0.79	2.71	1.53	9.6 × 10^5^
K0.5	10^−4^	1.43	1.43	0.08	8.2 × 10^4^
	10^−3^	1.69	1.69	0.02	9.1 × 10^4^
	10^−2^	1.55	1.55	0.1	1.2 × 10^5^
K0.5S1	10^−4^	1.98	2.31	0.32	3.3 × 10^5^
	10^−3^	2.37	2.37	0.07	1.8 × 10^5^
	10^−2^	2.06	2.06	0.23	4.7 × 10^5^
K0.5S2	10^−4^	2.45	2.80	0.55	5.7 × 10^5^
	10^−3^	3.19	3.19	0.65	1.0 × 10^6^
	10^−2^	3.37	3.37	0.60	1.1 × 10^6^
K1	10^−4^	2.05	2.05	0.05	5.7 × 10^4^
	10^−3^	2.56	2.56	0.14	1.4 × 10^5^
	10^−2^	2.50	2.50	0.12	6.3 × 10^4^
K1S1	10^−4^	2.22	2.44	0.17	2.2 × 10^5^
	10^−3^	2.91	2.91	0.35	5.4 × 10^5^
	10^−2^	3.32	3.32	0.34	3.4 × 10^5^
K1S2	10^−4^	3.12	3.12	0.45	1.4 × 10^6^
	10^−3^	4.33	4.33	0.64	1.5 × 10^6^
	10^−2^	4.28	4.28	0.76	1.1 × 10^6^
K1.5	10^−4^	2.45	2.45	0.13	2.4 × 10^5^
	10^−3^	2.92	2.92	0.10	1.7 × 10^5^
	10^−2^	2.91	2.91	0.09	1.9 × 10^5^
K1.5S1	10^−4^	3.26	3.26	1.24	1.9 × 10^6^
	10^−3^	3.65	3.65	0.25	5.7 × 10^5^
	10^−2^	4.38	4.38	1.06	1.8 × 10^6^
K1.5S2	10^−4^	4.93	4.93	0.64	8.7 × 10^5^
	10^−3^	5.09	5.09	0.77	9.4 × 10^5^
	10^−2^	6.30	6.30	0.91	9.0 × 10^5^
K2	10^−4^	3.02	3.02	0.10	1.0 × 10^5^
	10^−3^	3.38	3.38	0.13	1.6 × 10^5^
	10^−2^	2.82	2.82	0.12	8.9 × 10^4^
K2S1	10^−4^	3.64	3.64	0.21	2.3 × 10^5^
	10^−3^	3.93	3.93	0.44	7.7 × 10^5^
	10^−2^	4.65	4.65	0.77	1.2 × 10^6^
K2S2	10^−4^	4.15	4.15	0.41	6.7 × 10^5^
	10^−3^	5.06	5.06	0.55	1.4 × 10^6^
	10^−2^	5.11	5.11	0.21	5.3 × 10^5^

## Data Availability

The raw/processed data required to reproduce these findings cannot be shared at this time as the data also forms part of an ongoing study.

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
