# Peer review of "Experimental Investigation on Dynamic Tensile Behaviors of Engineered Cementitious Composites Reinforced with Steel Grid and Fibers"

_materials, 2021, doi:10.3390/ma14227042_

Round 1

Reviewer 1 Report

Generally my opinion is good.

But in the text are same technical erros. See in attched file. I think that Authors should used my suggestions. Than the paper it will be more easy to read.

Other problems are in references. The Autjors are from China and many references are from Chine. It is necessery to add same papers from others countries for examle form Europe as:

Vantadori Sabrina, Carpinteri Andrea, Głowacka Karolina, Greco Fabrizio, Osiecki Tomasz, Ronchei Camilla , Zanichelli Andrea: Fracture toughness characterisation of a glass fibre‐reinforced plastic composite, Fatigue & Fracture of Engineering Materials & Structures, vol. 44, nr 1, 2020, s. 3-13

and many others

Author Response

Manuscript ID: materials-1433967

Title: Experimental Investigation on Dynamic Tensile Behaviors of Engineered Cementitious Composites Reinforced with Steel Grid and Fibers

Authors: Liang Li, Hongwei Wang, Jun Wu, Shutao Li, Wenjie Wu

Materials

Dear Ms. Ruby. Ma

We would like to express our gratitude to the three reviewers for their time and efforts to our manuscript. The reviewers’ comments contribute to a significant improvement of the academic level of our manuscript.

We have read the reviewers’ comments, and carefully revised the manuscript. The manuscript has been revised in close accordance with their comments and suggestions. You can find the modified and revised parts which are marked in red in the manuscript. In addition, a one-by-one response to the three reviewers’ comments has been attached separately.

We are looking forward to further information on this manuscript.

Yours sincerely,

Liang Li, Shutao Li  (corresponding author)

Professor

Key Laboratory of Urban Security and Disaster Engineering, Beijing University of Technology, Ministry of Education, Beijing 100124, China

Institute of Defense Engineering, AMS, PLA, Beijing 100036, China

Email: liliang@bjut.edu.cn, list16@mails.tsinghua.edu.cn

6th November 2021

Reply to reviewer 1

Generally my opinion is good.

But in the text are same technical erros. See in attched file. I think that Authors should used my suggestions. Than the paper it will be more easy to read.

Other problems are in references. The Autjors are from China and many references are from Chine. It is necessery to add same papers from others countries for examle form Europe as:

Vantadori Sabrina, Carpinteri Andrea, Głowacka Karolina, Greco Fabrizio, Osiecki Tomasz, Ronchei Camilla , Zanichelli Andrea: Fracture toughness characterisation of a glass fibre‐reinforced plastic composite, Fatigue & Fracture of Engineering Materials & Structures, vol. 44, nr 1, 2020, s.3-13 and many others.

Reply:

We would like to express our gratitude to your time and efforts to our manuscript. The manuscript has been revised in close accordance with your comments and suggestions. The technical errors you mentioned all have been revised. You can find the modified and revised parts which are marked in red in the revised manuscript. On the other hand, we have added 8 papers from others countries as references. Articles are cited in Line 56 and the reference number is [33], Line 58 and the reference number is [34], Line 59 and the reference number is [35], Line 91 and the reference number is [53], Line 96 and the reference number is [58], Line 104 and the reference number is [64], Line 106 and the references number are [66-67].

Reference:

[33] Mechtcherine, V.; Michel, A.; Liebscher, M.; Schneider, K.; Großmann, C. Mineral impregnated carbon fiber composites as novel reinforcement for concrete construction: Material and automation perspectives, Automation in Construction. 2020, 110, 103002, doi: 10.1016/j.autcon.2019.103002

[34] Vantadori, S.; Carpinteri, A.; Głowacka, K.; Greco, F.; Osiecki, T.; Ronchei, C.; Zanichelli, A. Fracture toughness characterisation of a glass fibre‐reinforced plastic composite, Fatigue & Fracture of Engineering Materials & Structures, 2020, 44, 1, 3-13. doi:10.1111/ffe.13309.

[35] Khandelwal, S.; Rhee, KY. Recent advances in basalt-fiber-reinforced composites: Tailoring the fiber-matrix interface, Composites Part B: Engineering. 2020, 192, 108011, doi: 10.1016/j.compositesb.2020.108011.

[53] Daskiran, M M.; Daskiran, E G.; Gencoglu, M. Mechanical and durability performance of textile reinforced cementitious composite panels, Constr. Build. Mater. 2020, 264,120224, doi: 10.1016/j.conbuildmat.2020.120224.

[58] John, S K.; Nadir, Y.; Girija, K.; Giriprasad, S. Tensile behaviour of glass fibre textile reinforced mortar with fine aggregate partially replaced by fly ash, Materials Today: Proceedings. 2020, 27, 144–149, doi: 10.1016/j.matpr.2019.09.135.

[64] Halvaei, M.; Jamshidi, M.; Latifi, M.; Ejtemaei, M. Experimental investigation and modelling of flexural properties of carbon textile reinforced concrete, Constr. Build. Mater. 2020, 262, 120877, doi: 10.1016/j.conbuildmat.2020.120877.

[66] Kalaimathi, R P T, Balaji S, A review paper on mechanical properties of flexural and impact test on textile reinforced engineered cementitious composites. Materials Today: Proceedings. 2021, 09, 443, doi:10.1016/ j.matpr.2021.09.443.

[67] Daskiran, E G.; Daskiran, M M.; Gencoglu, M. Experimental investigation on impact strength of AR Glass, Basalt and PVA textile reinforced cementitious composites, European Journal of Environmental and Civil Engineering. 2020, 1-20, doi: 10.1080/ 19648189.2020.1749940.

Reviewer 2 Report

Summary

The paper by Liang Li et al. presents an experimental investigation on dynamic tensile behavior of concrete with PVA and KEVLAR fibers. A parametric study on fiber content is done for both fiber types. Samples are reinforced with 0, 1 and 2 steel mesh grids. Traction tests at three different loading rates are run for all samples until failure, maximum stress and maximum elongation are retained to extract conclusions. The manuscript concludes that that the optimal fiber concentration is 1.5% for both polyvinyl alcohol and KEVLAR, a higher fiber content results in fiber agglomeration which reduces the final strength of the samples. The study also concludes that PVA is best for deformation and energy dissipation while KEVLAR is best for maximum tensile strength.

Broad comments

The reviewer thanks the authors for the well conducted and presented research. Some comments can be found in the following:

  1. Can you specify how the three loading rates are determined? Are they representative of some real loading scenario?
  2. I am not sure whether or not it is stated in the manuscript, how many samples in total have been tested? 55? You can add this value in the caption of figure 3.
  3. Is there an estimation of the experimental results standard deviation, or some variability indicator? Only one test is being used in each case and conclusions are drawn from them without a statistical analysis. How significant are these results? For instance, case K2 shows a much lower peak than K1.5 in figure 22, same case for final elongation of K1.5 in figure 24, are these results significative?

Specific comments

Format tables without double spacing for a more compact presentation

Line 155: plastic sheets was covered on it --> was covered with plastic sheets

Figure 3 shows the ended samples before the preparation procedure has been introduced (including Figure 4). Better to place figure 3 at the end of the description and after figure 4.

Line 195: Revise sentence syntax

Line 196: Did you mean microscopic?

Line 2017 gained, --> obtained

Line 226: “…of the above specimens.” It seems that the sentence is incomplete, did you mean of the above specimens with 0, 1 and 2 steel grid layers?

Line 230-231: “appears a longer stress plateau” revise syntax

Figure 17: Matrix markers not clearly visible in the plot.

Line 346: “will be enhanced” --> is enhanced

Author Response

Manuscript ID: materials-1433967

Title: Experimental Investigation on Dynamic Tensile Behaviors of Engineered Cementitious Composites Reinforced with Steel Grid and Fibers

Authors: Liang Li, Hongwei Wang, Jun Wu, Shutao Li, Wenjie Wu

Materials

Dear Ms. Ruby. Ma

We would like to express our gratitude to the three reviewers for their time and efforts to our manuscript. The reviewers’ comments contribute to a significant improvement of the academic level of our manuscript.

We have read the reviewers’ comments, and carefully revised the manuscript. The manuscript has been revised in close accordance with their comments and suggestions. You can find the modified and revised parts which are marked in red in the manuscript. In addition, a one-by-one response to the three reviewers’ comments has been attached separately.

We are looking forward to further information on this manuscript.

Yours sincerely,

Liang Li, Shutao Li  (corresponding author)

Professor

Key Laboratory of Urban Security and Disaster Engineering, Beijing University of Technology, Ministry of Education, Beijing 100124, China

Institute of Defense Engineering, AMS, PLA, Beijing 100036, China

Email: liliang@bjut.edu.cn, list16@mails.tsinghua.edu.cn

6th November 2021

Reply to reviewer 2

The paper by Liang Li et al. presents an experimental investigation on dynamic tensile behavior of concrete with PVA and KEVLAR fibers. A parametric study on fiber content is done for both fiber types. Samples are reinforced with 0, 1 and 2 steel mesh grids. Traction tests at three different loading rates are run for all samples until failure, maximum stress and maximum elongation are retained to extract conclusions. The manuscript concludes that that the optimal fiber concentration is 1.5% for both polyvinyl alcohol and KEVLAR, a higher fiber content results in fiber agglomeration which reduces the final strength of the samples. The study also concludes that PVA is best for deformation and energy dissipation while KEVLAR is best for maximum tensile strength.

Reply:

We would like to express our gratitude to your time and efforts to our manuscript. The manuscript has been revised in close accordance with your comments and suggestions. You can find the modified and revised parts which are marked in red text in the revised manuscript. In addition, a one-by-one response to your comments has been attached separately. 

Comments 1

Broad comments

The reviewer thanks the authors for the well conducted and presented research. Some comments can be found in the following:

Can you specify how the three loading rates are determined? Are they representative of some real loading scenario?

Reply:

We appreciate the reviewer’s comment.

The three loading rates are determined according to the application of steel grid-fiber reinforced ECC studied in this paper. Understanding the tensile behaviors of ECC with different strain rates is vital to design the airport runway under aircraft taxiing and earthquake loading. The strain rate for aircraft taxiing and landing is about 10-2-1 s-1, and for earthquake loading is about 10-3-10-1 s-1. Therefore, the strain rates in this paper are determined as 10-4s-1, 10-3s-1 and 10-2s-1.

Comments 2

I am not sure whether or not it is stated in the manuscript, how many samples in total have been tested? 55? You can add this value in the caption of figure 3.

Reply:

Thanks for the reviewer’s advices. Actually, 135 samples in total have been tested in the current study. We have added this value in the context of the revised manuscript. 

Comments 3

Is there an estimation of the experimental results standard deviation, or some variability indicator? Only one test is being used in each case and conclusions are drawn from them without a statistical analysis. How significant are these results? For instance, case K2 shows a much lower peak than K1.5 in figure 22, same case for final elongation of K1.5 in figure 24, are these results significative?

Reply:

Thanks for the reviewer’s good question. And we are so sorry for not describing this procedure clearly.In the process of the test, each loading case has three specimens. The authors selected the experimental results closest to the average value of the three specimens for comparative analysis. The specimen of case K2 at the strain rate 10-4s-1 is damaged at the end of the specimen due to the stress concentration during the test. Therefore, case K2 shows a much lower peak stress than K1.5 in figure 22. Figure 24 shows that the elongation of the specimens is not in positive proportion to the fiber volume content, and the optimum content is 1.5%. 

Comments 4

Specific comments

Format tables without double spacing for a more compact presentation

Reply:

Thanks for the reviewer’s kind comments.

The authors had revised the table format of the revised manuscript.

Comments 5

Line 155: plastic sheets was covered on it --> was covered with plastic sheets.

Reply:

Thanks for the reviewer’s kind comments.

The authors had revised this content in line172 of the revised manuscript.

Comments 6

Figure 3 shows the ended samples before the preparation procedure has been introduced (including Figure 4). Better to place figure 3 at the end of the description and after figure 4.

Reply:

Thanks for the reviewer’s kind comments.

The authors had placed figure 3 at the end of the description and after figure 4.

Comments 7

Line 195: Revise sentence syntax

Reply:

Thanks for the reviewer’s kind comments.

The authors had revised this content in line216-217 of the revised manuscript.

Comments 8

Line 196: Did you mean microscopic?

Reply:

Thanks for the reviewer’s good question.

The authors used the electron microscope in this test and explained it in line 219-223 of the revised manuscript.

Comments 9

Line 2017 gained, --> obtained

Reply:

Thanks for the reviewer’s kind comments.

The authors had revised this content in line240 of the revised manuscript.

Comments 10

Line 226: “…of the above specimens.” It seems that the sentence is incomplete, did you mean of the above specimens with 0, 1 and 2 steel grid layers?

Reply:

Thanks for the reviewer’s good question.

The authors had revised this content in line249 of the revised manuscript.

Comments 11

Line 230-231: “appears a longer stress plateau” revise syntax

Reply:

Thanks for the reviewer’s kind comments.

The authors had revised this content in line254 of the revised manuscript.

Comments 12

Figure 17: Matrix markers not clearly visible in the plot.

Reply:

Thanks for the reviewer’s kind comments.

The authors had revised this content in Figure 17 of the revised manuscript.

Comments 13

Line 346: “will be enhanced” --> is enhanced

Reply:

Thanks for the reviewer’s kind comments.

The authors had revised this content in Line 371 of the revised manuscript.

Reviewer 3 Report

Dear Authors,

Please see general and specific comments in the attached file. A thorough English language review of the manuscript is recommended.

Author Response

Manuscript ID: materials-1433967

Title: Experimental Investigation on Dynamic Tensile Behaviors of Engineered Cementitious Composites Reinforced with Steel Grid and Fibers

Authors: Liang Li, Hongwei Wang, Jun Wu, Shutao Li, Wenjie Wu

Materials

Dear Ms. Ruby. Ma

We would like to express our gratitude to the three reviewers for their time and efforts to our manuscript. The reviewers’ comments contribute to a significant improvement of the academic level of our manuscript.

We have read the reviewers’ comments, and carefully revised the manuscript. The manuscript has been revised in close accordance with their comments and suggestions. You can find the modified and revised parts which are marked in red in the manuscript. In addition, a one-by-one response to the three reviewers’ comments has been attached separately.

We are looking forward to further information on this manuscript.

Yours sincerely,

Liang Li, Shutao Li  (corresponding author)

Professor

Key Laboratory of Urban Security and Disaster Engineering, Beijing University of Technology, Ministry of Education, Beijing 100124, China

Institute of Defense Engineering, AMS, PLA, Beijing 100036, China

Email: liliang@bjut.edu.cn, list16@mails.tsinghua.edu.cn

6th November 2021

Reply to reviewer 3

Dear Authors,

Please see general and specific comments in the attached file. A thorough English language review of the manuscript is recommended.

Reply:

We would like to express our gratitude to your time and efforts to our manuscript. The manuscript has been revised in close accordance with your comments and suggestions. You can find the modified and revised parts which are marked in red text in the revised manuscript. In addition, a one-by-one response to your comments has been attached separately.

Comments 1

The Abstract should be rewritten based on the comments. Abstract should be clear enough to give a general concept of the research. The Abstract also lacks figures of the result. It is recommended that the authors state some obtained results.

Reply:

Thanks for the reviewer’s kind comments.

The abstract has been rewritten based on the comments. Some obtained results have been added in the abstract in line 23-26 and line 29-32.

Comments 2

Line 15: What was used to reinforce the ECC?

Reply:

Thanks for the reviewer’s good question.

The ECC was reinforced with the steel grids and fibers. The authors had added this content in line 14 of the revised manuscript.

Comments 3

Line 18: State the meaning before the acronym at first mention,

Reply:

Thanks for the reviewer’s kind comments.

The authors had revised this content in line17 of the revised manuscript.

Comments 4

Line 38-39: These citations should rather be presented as [5-12]. There is no point listing them out without specific statements.

Reply:

Thanks for the reviewer’s kind comments.

The authors had revised this content in line 45 of the revised manuscript.

Comments 5

Line 62: The unit expression is not correct

Reply:

Thanks for the reviewer’s kind comments.

The authors had revised this content in line 70 of the revised manuscript.

Comments 6

Line 76: What is DIF?

Reply:

Thanks for the reviewer’s kind comments.

DIF is the dynamic increasing factor. It is the ratio of dynamic test results to corresponding static results. The authors had added this content in line 85 of the revised manuscript.

Comments 7

Line 77: Acronyms used in the text should be written out at first mention

Reply:

Thanks for the reviewer’s kind comments.

The authors had revised this content in line 86 of the revised manuscript.

Comments 8

Line 88-90: Rewrite this statement and change the citation as suggested above

Reply:

Thanks for the reviewer’s kind comments.

The authors had rewritten this statement and change the citation as suggested by the reviewer in line 98 of the revised manuscript.

Comments 9

Line 118: The chemical property of cement is an important aspect which has been omitted. Strength development in ECC is a function of formation of calcium-silicate hydrates.

Reply:

Thanks for the reviewer’s kind comments.

The chemical property of cement had been added in Table 2.

Comments 10

Line 132: What are these fractions based on? Is it cement or ECC matrix?

Reply:

Thanks for the reviewer’s kind comments.

The volume fraction of these two types fibers are 0%, 0.5%, 1%, 1.5%, 2% of ECC matrix. The authors had clarified this content in line 148 of the revised manuscript.

Comments 11

Line 137: This is not clear. What is the mass ratio based on?

Reply:

Thanks for the reviewer’s good question.

In this mix proportions, other components are the mass ratio to cement. The authors had clarified this content in line 145 of the revised manuscript.

Comments 12

Line 164: Does the procedure follow a specific standard or publication? If yes, it should be stated accordingly

Reply:

Thanks for the reviewer’s good question.This procedure for the preparation of tested specimens is adopted mainly to ensure the ECC specimen components mixed adequately and fibers spread in the ECC matrix uniformly. It comes from the repeating trial conducted by the authors before tensile loading tests.

Comments 13

Line 175-178: This paragraph should be rewritten, e.g., M represents what? (is it control?) A represents PVA, K represents KEVLAR at different fiber contents between 0.5 and 2%. S represents the steel grid at different layer numbers between 1 and 2.

Reply:

Thanks for the reviewer’s kind comments.

The authors had rewritten this content in line 192-196 of the revised manuscript.

Comments 14

Line 186: Why did authors choose this strain rate?

Reply:

Thanks for the reviewer’s good question.Due to the large number of specimens, it is unrealistic to show each group of specimens. We selected the specimen failure mode corresponding to the middle value (10-3s-1) of three strain rates (10-4s-1, 10-3s-1, 10-2s-1) for comparative analysis.

Comments 15

Line 196: Do the authors refer to microscopic?

Reply:

Thanks for the reviewer’s good question.

The cross-section microcosmic view of damaged steel grid-PVA ECC specimen (=10-3s-1) was obtained by scanning electron microscope.

Comments 16

Line 213: This Figure would be better explained if it was labeled and the mechanism identified

Reply:

Thanks for the reviewer’s kind comments.The authors have added the corresponding strain rate of figure 8 in the figure caption. The mechanism related to the fiber bonding action revealed by this figure has been expressed in line291-223 of the revised manuscript.

Comments 17

Line 221: Reformat this Table such that a specimen type begins on a new set of result line (i.e. starting with strain rate 10-4)

Reply:

Thanks for the reviewer’s good suggestion.

This table has been reformatted such that a specimen type begins on a new set of result line.

Comments 18

Table7: Correct M to C

Reply:

Thanks for the reviewer’s kind comments.

In the denotation of specimen type, “M” stands for the matrix specimen without addition of steel grid and fibers. “M” is the initials of “matrix”. The authors think that it is reasonable to use “M” stands for the matrix specimen.

Comments 19

Line 227: Change ECC matrix to control where applicable

Reply:

Thanks for the reviewer’s kind comments.

In the current test study, the ECC matrix specimens without addition of steel grid and fibers were used as the benchmark for the comparative analysis.

Comments 20

Figure10: Matrix can be changed to control (C)

Reply:

Thanks for the reviewer’s kind comments.

In this figure, “Matrix” stands for the matrix specimen without addition of steel grid and fibers. In the current test study, the matrix specimen was used as the benchmark for the comparative analysis.

Comments 21

Line 241: Why was these groups specifically chosen?

Reply:

Thanks for the reviewer’s good question.

The authors found that the specimens in the group of A0.5 and A1 had cracked in the central part of the specimen. This is an ideal failure mode. So the specimens in the group of A0.5 and A1 can accurately reflect the effect of steel grid on the tensile behaviors of the ECC specimen.

Comments 22

Figure16: Again, it would be necessary to identify any mechanism observed directly in the micrographs

Reply:

Thanks for the reviewer’s kind comments.

As mentioned in the section 3.2.1, the brittle failure pattern can be observed for the steel grid-KEVLAR-ECC specimens at strain rate of 10-3, and the multi-cracks phenomenon cannot be observed obviously. The mechanism for this phenomenon can be explained by the micrographs in figure 16. It is found in the micrographs that KEVLAR fibers are unevenly distributed in the ECC matrix and some fiber agglomerations can be observed. This has been expressed in line308-317 of the revised manuscript. The reason for the fiber agglomerating is also discussed in the revised manuscript.

Comments 23

Line 309: Reformat as in Table 7. The results obtained for the control experiments were supposed to be the same for both Tables 7 and 8. Check and correct where applicable

Reply:

Thanks for the reviewer’s good suggestion.

This table has been reformatted according to the reviewer’s suggestion. In the dynamic tensile tests of steel grid-PVA fiber and KEVLAR fiber ECC, the matrix specimen without addition of steel grid and fibers was all used as the benchmark for the comparative analysis.

Comments 24

Table 8: Change M to C as earlier advised

Reply:

Thanks for the reviewer’s kind comments.

In the denotation of specimen type, “M” stands for the matrix specimen without addition of steel grid and fibers. “M” is the initials of “matrix”. The authors think that it is reasonable to use “M” stands for the matrix specimen.

Comments 25

Line 323: Why were these groups chosen and not the higher fiber contents?

Reply:

Thanks for the reviewer’s good question.

This section analyzes the effect of the number of steel grid layer on the tensile behaviors of steel grid-KEVLAR fiber ECC. The authors found that the specimens in the group of K0.5 and K1 had cracked in the central part of the specimen. This is an ideal failure mode. So the specimens in the group of K0.5 and K1 can accurately reflect the effect of steel grid on the tensile behaviors of the ECC specimen.

Comments 26

Line 326, 328: without steel grid

Reply:

Thanks for the reviewer’s kind comments.

The authors had revised this content in line 352 and line354 of the revised manuscript.

Comments 27

Line 361: at different strain rates

Reply:

Thanks for the reviewer’s kind comments.

The authors had revised this content in line 387 of the revised manuscript.

Round 2

Reviewer 2 Report

2nd round review comments:

In the response to broad comment “3” the authors state that three specimens are tested for each loading case and the case with closest value to the average retained. I cannot find a mention to this in the revised version of the manuscript. Some questions arise: what average is used to pick the closest case? Peak strength or final elongation?  What are the minimum and maximum values among the three samples in each case? Can we assume that the three samples follow a normal distribution in order to use the average metrics?

A statistical analysis section should be included answering all those questions and the full data of all the tests included either in an annex, supplementary data or externally hosted repository in order to provide soundness and significance to the study.

The abstract is now far above the maximum of 200 words set by MDPI. I know that the inclusion of more results follows the recommendation of some of the reviewers, but a too lengthy and complex abstract undermines its readership attraction strength.

Other remarks have been correctly addressed in the previous revision.

Author Response

Manuscript ID: materials-1433967

Title: Experimental Investigation on Dynamic Tensile Behaviors of Engineered Cementitious Composites Reinforced with Steel Grid and Fibers

Authors: Liang Li, Hongwei Wang, Jun Wu, Shutao Li, Wenjie Wu

Materials

Dear Ms. Ruby. Ma

We would like to express our gratitude to the three reviewers for their time and efforts to our manuscript. The reviewers’ comments contribute to a significant improvement of the academic level of our manuscript.

We have read the reviewers’ comments, and carefully revised the manuscript. The manuscript has been revised in close accordance with their comments and suggestions. You can find the modified and revised parts which are marked in red in the manuscript. In addition, a one-by-one response to the three reviewers’ comments has been attached separately.

We are looking forward to further information on this manuscript.

Yours sincerely,

Liang Li, Shutao Li  (corresponding author)

Professor

Key Laboratory of Urban Security and Disaster Engineering, Beijing University of Technology, Ministry of Education, Beijing 100124, China

Institute of Defense Engineering, AMS, PLA, Beijing 100036, China

Email: liliang@bjut.edu.cn, list16@mails.tsinghua.edu.cn

13th November 2021

Reply to reviewer 2

We would like to express our gratitude to your time and efforts to our manuscript. The manuscript has been revised in close accordance with your comments and suggestions. You can find the modified and revised parts which are marked in red text in the revised manuscript. In addition, a one-by-one response to your comments has been attached separately.

Comments 1

In the response to broad comment “3” the authors state that three specimens are tested for each loading case and the case with closest value to the average retained. I cannot find a mention to this in the revised version of the manuscript. Some questions arise: what average is used to pick the closest case? Peak strength or final elongation?  What are the minimum and maximum values among the three samples in each case? Can we assume that the three samples follow a normal distribution in order to use the average metrics?

A statistical analysis section should be included answering all those questions and the full data of all the tests included either in an annex, supplementary data or externally hosted repository in order to provide soundness and significance to the study.

Reply:

Thanks for the reviewer’s good comments. We are very sorry for the inexact description about the test method in the previous revision. Actually, in the current test study, four specimens were tested for every loading case, i.e. every combination of certain fiber volume content, layer number of steel grid and tensile strain rate, and the average value of the test results was used for the comparative analysis. This description has been added to the revised manuscript. 180 PVA-ECC specimens and 180 KEVLAR-ECC specimens in total have been tested in the current study. These values are presented in the revised manuscript.

In fact, due to the limited number of tested specimens for every loading case, we cannot confirm that the test results of four specimens follow a normal distribution. The main aim of current test study is to investigate the effect of fiber volume content, layer number of steel grid and tensile strain rate to the tensile behaviors of the steel grid-fiber reinforced ECC. So we follow the usual convention and use the average value of the test results of every loading case to conduct the comparative analysis. We comprehensively considered the average results of peak strength and ultimate strain. We think that the averaged test results can characterize the effecting behaviors of the concerned factors on the whole.

We are also very sorry for not providing the statistical analysis and full data of the test results due to the consideration of the manuscript length. And these data are also forms part of an ongoing study

Comments 2

The abstract is now far above the maximum of 200 words set by MDPI. I know that the inclusion of more results follows the recommendation of some of the reviewers, but a too lengthy and complex abstract undermines its readership attraction strength.

Reply:

Thanks for the reviewer’s kind advices. The abstract has been reduced somewhat according to the reviewer’s advices.

Reviewer 3 Report

Dear Authors,

You may modify the Abstract with these sentences before the results

"The mixture was designed with different volume fractions of fibers and layer numbers of steel grids to explore the reinforcement effectiveness on the dynamic performance of ECC. The volume fractions of these two types of fibers were 0%, 0.5%, 1%, 1.5%, 2% of ECC matrix, respectively. The layer number of steel grid was 0, 1, 2."

Author Response

Reply to reviewer 3

We would like to express our gratitude to your time and efforts to our manuscript. The manuscript has been revised in close accordance with your comments and suggestions. You can find the modified and revised parts which are marked in red text in the revised manuscript. In addition, a one-by-one response to your comments has been attached separately.

Comments 1

Dear Authors,

You may modify the Abstract with these sentences before the results

"The mixture was designed with different volume fractions of fibers and layer numbers of steel grids to explore the reinforcement effectiveness on the dynamic performance of ECC. The volume fractions of these two types of fibers were 0%, 0.5%, 1%, 1.5%, 2% of ECC matrix, respectively. The layer number of steel grid was 0, 1, 2.".

Reply:

Thanks for the reviewer’s kind comments.

The abstract has been revised according to the reviewer’s comments.